# Genetic and Epigenetic Regulation of *Drosophila* Oocyte Determination

**DOI:** 10.3390/jdb11020021

**Published:** 2023-05-24

**Authors:** Brigite Cabrita, Rui Gonçalo Martinho

**Affiliations:** Department of Medical Sciences, Institute of Biomedicine (iBiMED), University of Aveiro, Agra do Crasto, Edifício 30, 3810-193 Aveiro, Portugal; b.cabrita@ua.pt

**Keywords:** *Drosophila*, oogenesis, oocyte determination, transcriptional regulation, translational regulation, Epigenetics

## Abstract

Primary oocyte determination occurs in many organisms within a germ line cyst, a multicellular structure composed of interconnected germ cells. However, the structure of the cyst is itself highly diverse, which raises intriguing questions about the benefits of this stereotypical multicellular environment for female gametogenesis. *Drosophila melanogaster* is a well-studied model for female gametogenesis, and numerous genes and pathways critical for the determination and differentiation of a viable female gamete have been identified. This review provides an up-to-date overview of *Drosophila* oocyte determination, with a particular emphasis on the mechanisms that regulate germ line gene expression.

## 1. Introduction

“A beginning is the time for taking the most delicate care that the balances are correct.”


*(From the book “Dune” by Frank Herbert)*


### Germ Line Cyst and Oocyte Determination

The polyploidization and formation of multinucleated structures are recurrent strategies to regulate cell size and function (reviewed in [1]). These strategies are observed in various organisms, where they play important roles in cell and tissue differentiation. This is accomplished by either forming nuclear collectives or by increasing the number of nuclear genome copies within a common cytoplasm. Cells with increased genome content can exist temporarily, for example, during the development of germ cells, or can be a permanent characteristic of mature somatic tissues, such as in the cardiac and skeletal muscles (reviewed in [1]). Primary oocyte determination occurs in many organisms within a germ line cyst, a multicellular structure composed of germ cells interconnected by intercellular bridges (reviewed in [2]). The underlying mechanisms behind oocyte determination and differentiation remain, however, not entirely understood, as the structure and organization of these cysts are surprisingly diverse.

Despite being in a cytoplasmic continuum, cells within the *Drosophila* and mouse germ line cysts differentiate into distinct cell types: one becoming the oocyte and the others becoming supporting nurse cells (reviewed in [3,4,5]) [6,7]. Once the presumptive oocyte is determined, the interconnected supporting cells will transfer their cytoplasmic components and organelles into it, ensuring correct differentiation of the oocyte (reviewed in [2]). However, in spite of significant similarities between *Drosophila* and mouse cysts, the contribution of the supporting nurse cells to primary oocyte maturation in these two organisms is distinct: *Drosophila* nurse cells become highly polyploid due endoreplication and die by programmed cell death at late stages of oocyte development, whereas the mouse nurse cells die early on during the process of primary oocyte determination without detectable changes of ploidy (reviewed in [2,8]) [6,7]. Mouse primary oocyte growth and maturation are then supported by somatic granulosa cells within the primordial and primary follicles (reviewed in [8]).

While meroistic insects (e.g., *Drosophila*, bees, butterflies, and mosquitoes) specify oocytes and supporting nurse cells within an interconnected germ line cyst, ancestral panoistic insects (e.g., crickets, grasshoppers, stoneflies, and thrips) specify oocytes without nurse cells, with or without the formation of cysts [9]. The question of whether the observed differences are attributable to overlooked or secondarily lost multicellular structures or if nurse cell differentiation emerged independently multiple times throughout insect evolution is still a matter of debate. Nevertheless, it is important to note that germ line cysts containing both oocytes and supporting cells have been observed in various invertebrate non-arthropod organisms (e.g., tardigrades [10]), and these multicellular structures were also described in distinct vertebrate organisms. For example, similarly to mice, *Xenopus*, Medaka, and zebrafish determine primary oocytes within multicellular germ line cysts [11,12,13], although it is still unclear if they differentiate supporting nurse cells.

The fact that primary oocyte specification within a multicellular structure is a conserved feature of female gametogenesis, yet the structure of the germ line cyst is itself highly diverse, raises intriguing questions about the benefits of this stereotypical multicellular environment for female gametogenesis, the underlying mechanisms driving cell fate decisions, and its potential impact on the evolution of distinct reproductive strategies.

## 2. *Drosophila* Melanogaster Oogenesis

Female gametogenesis has been particularly well studied in *Drosophila melanogaster*, with many seminal contributions to the field (reviewed in [3,4,5,14,15]). The *Drosophila* ovary, two per female, contains approximately 16–20 ovarioles. Each ovariole is composed of two distinct regions: the anterior localized germarium and the vitellarium, which corresponds to a linear array of progressively older egg chambers. *Drosophila* oogenesis starts in the germarium with an asymmetric division of an anteriorly localized germ line stem cell (GSC) and the formation of a cystoblast daughter cell (Figure 1) (reviewed in [16]). The germ line stem niche is composed of somatic terminal filament cells and cap cells that anchor the GSCs to the niche via adherens junctions and gap junctions and whose paracrine function is critical for the maintenance of the stem cell pool.

GSCs homeostasis and the correct differentiation of cystoblasts ensure an efficient production pipeline of oocytes without the formation of stem cell tumors (reviewed in [16]). GSCs’ mitotic divisions are typically asymmetric and orthogonal to the niche in such a way that the anteriorly localized daughter cell stays at the stem cell niche and maintains an undifferentiated stem cell fate, whereas the posterior localized daughter cell has the ability to exit the niche and initiate a highly complex differentiation program.

Briefly, somatic Cap cells secrete two bone morphogenetic protein (BMP) ligands (Decapentaplegic (Dpp) and Glass bottom boat (Gbb)), which are sequestered by GSCs and promote their self-renewal and inhibit differentiation (Figure 2) [17,18,19]. A critical effector of GSCs maintenance is the transcription factor Mothers against Dpp (Mad), which is translocated to the nucleus after being phosphorylated by activated BMP receptors (Figure 3) [17,20]. Among other targets, pMad inhibits transcription of the differentiation factor bag-of-marbles (bam) [18,20]. Remarkably, and clearly suggesting a pivotal role of these proteins, *bam* mutants or increased levels of Dpp signaling is associated with stem cell/pre-cystoblasts tumors in the germarium due to the inability to initiate differentiation, whereas reduced Dpp signaling or bam overexpression is associated to loss of the stem cell pool due to precocious differentiation of the GSCs [17,20]. It should be noted that the intermingled somatic escort cells also have a critical role in the regulation of GSC self-renewal, cystoblast mitotic divisions, and cystocyte (also known as cyst cells) differentiation, as they secrete distinct signaling molecules (e.g., Hedgehog (Hh), Epidermal Growth Factor (EGF), Wingless (Wg), insulin, and ecdysone) whose function is crucial for the regulation of early oogenesis (Figure 2) [21,22,23,24,25,26,27,28].

In well-fed young females, the GSCs divide approximately every 12–14 h [29], with a short G1 and M phase and a long G2 phase [30,31]. After CycB-dependent abscission from the GSC [32], the recently-divided cystoblast undergoes four rounds of incomplete mitotic divisions to produce a cyst of 16 interconnected cells. Cyclin/Cdk protein levels are high during the initial cyst divisions, but they decrease as the cyst reaches the last round of divisions [33], fully supporting the hypothesis that Cyclin/Cdk protein levels regulate the number of mitotic rounds of the cystocytes, overexpression of CycA and CycB is associated with an extra round of mitotic divisions and the formation of 32-cell cysts [34].

### 2.1. Cystoblast, Fusome, and Germ Line Cyst Formation

As mentioned before, once formed, the cystoblast goes through four rounds of mitotic division with incomplete cytokinesis. This gives rise to a cyst of 16 cells interconnected through modified cytoplasmic bridges known as ring canals [35,36]. The pattern of the cystocytes divisions is highly stereotypical, and the two initial daughter cells are invariably at the center of the 16-cell cyst and contain four-ring canals (reviewed in [3]). Whereas one of these two cystocytes (also known as pro-oocytes) will differentiate into an oocyte, the remaining 15 cells will differentiate into supporting nurse cells. Interestingly, the remaining pro-oocyte and two of the cystocytes with three-ring canals can transiently enter meiosis before exiting meiosis and progressing into a nurse cell differentiation program [37,38], which suggests a sequential cell fate decision process.

The fusome is a large, highly dynamic cytoplasmic structure that extends through the ring canals to all cells of the cyst and whose function is critical for their symmetric division and for oocyte determination (reviewed in [39]) [40]. Fusome molecular composition is similar to the spectrosome—a rounded membranous cytoskeleton structure present in the GSCs. Initially, a small plug of fusome material can be observed in the differentiating pre-cystoblast. Subsequently, during each division, the fusome is nucleated after the disassembly of the mitotic spindle, eventually giving rise to a continuous, branched structure [40]. The fusome serves multiple functions within the cyst, including synchronization of mitotic divisions, anchoring the cystocytes’ centrosomes, orienting the planes of mitotic divisions, and facilitating cyst symmetry breaking, polarization of the microtubules and oocyte determination [40,41,42,43]. The fusome is composed of distinct membrane cytoskeleton proteins, including Hu-li tai shao (Hts), Ankyrin, and Spectrins [41,44,45], potentially explaining why this structure synchronizes cystocytes development, the fusome membrane vesicles/tubules are continuous across all cells of the cyst, and they connect to the cytoplasmic endoplasmic reticulum (ER) of each cell [46]. The fusome also contains components of the recycling endosomal and lysosomal compartments [47]. Although many components of the fusome are apparently dispensable, Rab11, a regulator of the recycling endosome, is required for GSC divisions and fusome integrity [47].

Hts, an Adducin-like protein, is an actin filament capping protein [48] that recruits Spectrin to actin filaments, promoting the formation of an extended Spectrin–Actin network. Hts is required for fusome formation, cyst mitotic divisions, ring canal formation and maturation, and oocyte determination [41,49]. This clearly demonstrates the critical role of this and other membrane skeletal proteins for cyst maturation, oocyte determination, and egg chamber formation. Interestingly, an alternatively spliced protein isoform of Hts, Ovhts (also known as hts-PA), is cleaved into two products whose fragments differentially localize to the fusome of mitotic dividing cystocytes or the germ cells ring canals cells [50]. Although with a limited impact on female fertility, the abnormal cleavage of the Ovhts protein isoform significantly delays fusome disassembly after the end of the cystocyte’s mitotic divisions [50]. Par-1, a protein kinase whose function is pivotal for cell polarization, also localizes to the fusome, and it is required for the maintenance of oocyte fate [51]. Mutants for *par-1* exhibit an anomalous arrangement of the cyst microtubules, which impairs the transport of oo18 RNA-binding protein (Orb) and centrosomes from the anterior to the posterior region of the oocyte [51].

### 2.2. First among Equals: Oocyte Determination

Polytrophic meroistic insects, like *Drosophila*, differentiate two cell types (oocyte and supporting nurse cells) from within the same germ line cyst (reviewed in [3]). This implies that regardless of the cytoplasmic continuum, the cystocytes are able to engage in distinct differentiation programs.

*Drosophila* primary oocyte determination requires cyst symmetry breaking, the polarization of the microtubule network, and dynein-dependent transport of determinants into the presumptive oocyte (reviewed in [3,15]). This results in the formation of a cyst containing one single oocyte and 15 nurse cells. Failure to break such symmetry or to perform such transport (e.g., mutations in the dynein-dynactin complex or incubation with microtubule-depolymerization drugs) are typically associated with the loss of the presumptive oocyte and differentiation of 16 nurse cells within the cyst (reviewed in [3,15]).

The oocyte is invariably determined from one of the two cystocytes with four-ring canals [37]. Two alternative models explain oocyte selection from the two four-ring canal pro-oocytes: one model proposes that the two pro-oocytes are originally indistinguishable and that oocyte selection only occurs after the cystocyte’s mitotic divisions. In region 2A of the germarium, which is the stage when the 16-cell cyst has already been formed, the two pro-oocytes and the two cystocytes with three-ring canals enter meiosis and form the synaptonemal complex (SC) [37,38]. Only the two pro-oocytes will remain in meiosis once cyst symmetry is broken and the cystocyte’s mitotic divisions are complete. This model proposes a stochastic selection of the oocyte after a competition between the two four-ring canal pro-oocytes [37]. The remaining pro-oocyte exits meiosis and reverts to a nurse cell fate. The second model suggests instead that oocyte selection occurs during the first mitotic division of the cystoblast, as the fusome is asymmetrically inherited and one of the pro-oocytes, the GSCs daughter cell, will have a larger amount of fusome when compared to the other pro-oocyte and the remaining cystocytes [40,42]. The cystocyte with the highest levels of fusome will accumulate oocyte determinants, receive the centrosomes, mitochondria, and Golgi apparatus of the remaining cystocytes, and become the oocyte [52,53].

In agreement with the second model, it was recently shown that the minus-end-stabilizing protein Patronin associates with the fusome and stabilizes minus-end microtubules [43]. The initial fusome asymmetry created during the first mitotic division of the cystoblast results in marginally increased levels of Patronin in the future oocyte, which further enhances the stabilization of minus-end microtubules and leads to a weak polarization of the cyst microtubules. Such microtubule polarization is subsequently expanded by the Dynein-dependent transport of Patronin to the minus-end microtubules, further stabilization of these ends in the presumptive oocyte, and further enhancement of dynein-dependent transport of oocyte determinants to this cell. This positive feedback loop allows the amplification of a weak symmetry-breaking event into a robust polarization of the cyst microtubules.

The observation that oocyte determination relies on a Patronin-dependent noncentrosomal microtubule organizing center (ncMTOC) is highly significant [43]. The fusome is required for the sequential migration of the cyst centrosomes into the future oocyte [52]. This migration is defective when the integrity of the fusome is impaired, but not after colcemid, a microtubule inhibitor, treatment, or in mutants known to be defective for the polarization of the cyst microtubules or dynein-dependent transport of oocyte determinants [52]. Consistently, centrosomes are inactive during their migration into the oocyte [52]. Interestingly, and although the presumptive oocyte centrosomes are likely to contribute to the polarization of the cyst microtubules, oocyte determination relies on Patronin and its ability to organize early on an MTOC within the oocyte and induce a polarized network of microtubules within the cyst [43].

### 2.3. Oocyte-Specific Factors

The process of oocyte determination is closely associated with entry into meiosis and the accumulation of distinct oocyte-specific factors. Some of these factors, such as Bicaudal D (BicD) and Egalitarian (Egl), are adaptor proteins that connect diverse cargos, such as proteins, mRNAs, and organelles, to the dynein/dynactin motor, for minus-end directed transport [54,55,56,57]. Others, for example, correspond to oocyte-specific RNA binding proteins, such as Orb, Oskar (Osk), and fs(2)Cup (Cup) [58,59,60], which are crucial for correct translational regulation of distinct mRNAs and germ plasm assembly. Orb gene mutations have distinct effects on cyst development: null alleles cause defects in the fourth mitotic division and formation of eight-cell cysts, while weaker alleles result in the formation of 16-cell cysts, where oocyte and nurse cell specification is initiated but ultimately fail due to abnormal differentiation of the oocyte [59]. This abnormal differentiation results from a disorganized cyst organization and reduced accumulation of oocyte-specific factors.

Orb is a cytoplasmic polyadenylation element binding protein (CPEB) that regulates the localization and translation of distinct oocyte-specific mRNAs [59,61,62,63,64]. Reinforcing the idea that the robustness of oocyte determination relies on distinct positive feedback loops within the germ line cyst, Orb positively regulates its own translation [60,65,66]. BicD and Egl adaptor proteins form a protein complex that interacts with the Dynein light chain and the minus-end directed microtubules dynein motor complex [57]. This is crucial for ensuring the correct localization of distinct oocyte determinants, including Orb mRNA and protein, within the oocyte. In *BicD* and *egl* mutants, oocyte determination is compromised, leading to the differentiation of a cyst with 16 nurse cells [37,38,55,59].

### 2.4. Shall I Stay or Shall I Go: Entry into Meiosis

Meiotic prophase I is dependent on several critical steps, including homologous chromosome pairing, synapsis formation, and proper formation and maturation of meiotic cross-overs (reviewed in [14,67]). The Synaptonemal complex (SC) is a proteinaceous scaffold that is assembled between the paired homologous chromosomes, and it is required to stabilize chromosome pairing interactions during meiosis I. In *Drosophila*, the pairing of homologous autosomal chromosomes occurs during the pre-meiotic mitotic divisions of the cystocytes [68]. Once the 16-cell cyst is formed, the paired centromeres cluster, on average, into two groups. The meiotic SC genes are already expressed in the mitotically dividing cystocytes, and their association with the centromeric region is required for the correct pairing and clustering of the centromeres [68,69].

Early on, four nuclei within the 16-cell cyst assemble the SC along their euchromatic arms, but as the cyst matures, the euchromatic SC disassembles from two, and then one of these cystocytes, and only a single oocyte maintains a full-length SC in region 3 of the germarium (Figure 1) [37,38]. Interestingly, mutants for *egl* and *orb*, but not for *BicD*, show a transient entry of all cystocytes into meiosis, with the detectable assembly of the SC, before they all exit meiosis and differentiate as nurse cells. The fact that multiple cystocytes within the 16-cell germ line cyst attempt to enter meiosis early on, before exiting meiosis and engaging into a nurse cells differentiation program, suggest that the mechanisms required for oocyte determination are likely to go hand-in-hand with a poorly understood restriction mechanism that avoids the formation of multiple oocytes within a single germ line cyst [37,38].

## 3. Germ Cells Differentiation: Regulation of Gene Expression

### 3.1. Post-Transcriptional Regulation

GSC self-renewal and differentiation of the mitotically-dividing germ cell precursors are regulated by distinct regulators of translation (Figure 3) (reviewed in [70]). The translational repressor Pumilio (Pum), a founding member of the PUF family of RNA binding proteins, in conjunction with Nanos, is required for GSC self-renewal as they repress the translation of several differentiation factors [71,72,73]. For example, by recruiting the CCR4-NOT deadenylase complex [74], Nanos and Pum repress translation of the differentiating factors Mei-P26 and Brain tumor (Brat) [74,75,76].

The transcription factor Mad is critical for GSC self-renewal, and after being phosphorylated by activated BMP receptors, it is translocated to the nucleus [17,20]. Among other targets, pMad inhibits the transcription of differentiation factor bam [18,20]. Reduced BMP-signaling in the differentiating cystoblast leads to transcription of bam [18,20]. Bam, and its obligate co-factor Benign gonial cell neoplasm (Bgcn), form a translational repressor complex that antagonizes Nanos expression [77]. Consistently, germ line expression of Nanos is rapidly downregulated in the cystoblast daughter cell, as nanos mRNA translation is cooperatively repressed by Sex-lethal (Sxl), Bam, Bgcn, and Mei-P26 [78,79].

The relationship between Pum, Nanos, and Brain tumor (Brat) is crucial for the switch from a GSC’s self-maintenance program to the onset of cystoblast differentiation [76]. Brat and Nanos compete for Pum binding, and the two translation repressor complexes Nanos-Pum and Brat-Pum target distinct subsets of mRNAs. Whereas the Nanos-Pum complex primarily represses the translation of distinct differentiation factors, such as Mei-P26 and Brat [74,75,76], Brat-Pum primarily represses the translation of GSC self-renewal factors, such as Mad and Myc [76]. This generates a positive feedback loop for germ cell differentiation, where reduced levels of Nanos attenuate the translation repression of brat mRNA, and Brat further competes with Nanos for Pum binding and enhances its own translation.

More recently, it was suggested that the differentiation of germ cell precursors is a progressive process. In the early stages, the differentiation of cystoblasts is triggered by various factors, including the expression of Bam. However, in the eight-cell cyst, a switch from mitotic to meiotic programming is induced by the expression of RNA-binding Fox protein 1 (Rbfox1) [80]. Rbfox1 expression increases during early germ cell differentiation, and this RNA-binding protein directly represses the translation of pum mRNA. Consistent with this hypothesis, loss of Rbfox1 and ectopic expression of pum blocks cystocytes differentiation and results in germ-line tumors due to cyst breakdown and dedifferentiation of cystocytes into single mitotically proliferating cells [80].

Bruno, also known as arrest, encodes a translational repressor whose function is required to maintain mitotic quiescence during meiosis [33]. Bruno represses the translation of mitotic cyclins during meiosis, and in the absence of this protein, there is an ectopic expression of mitotic cyclins during meiosis. This results in the inhibition of cystocyte differentiation and the generation of mitotically active germ cells, with complete cytokinesis and expression of high levels of cytoplasmic Bam and Sxl (early cystoblast markers) but that do not accumulate markers of late cystocytes or differentiating oocytes.

### 3.2. Transcriptional Regulation

The epigenome of the *Drosophila* female germ line is very dynamic, with high levels of positive and negative marks [81]. Not surprisingly, distinct epigenetic modifications and chromatin remodeling proteins have been reported to be required for the maintenance of the GSC, cystoblast, cystocyte differentiation, and oocyte determination (Figure 4) (reviewed in [82]).

#### 3.2.1. Repressive Histone Marks (H3K9me3 and H3K27me3)

The Polycomb system consists of two multi-protein complexes, namely Polycomb repressive complex 1 (PRC1) and Polycomb repressive complex 2 (PRC2) (reviewed in [83,84,85]). PRC1 functions as an E3 ubiquitin ligase that catalyzes the monoubiquitylation of histone H2A at lysine 119, resulting in the formation of H2AK119ub1 (reviewed in [83]). PRC2, on the other hand, is a histone methyltransferase that is capable of mono-, di-, or trimethylating histone H3 at lysine 27, leading to the formation of histone H3 lysine 27 mono-, di-, and trimethylation (H3K27me1/2/3). PRC1 and PRC2 function by binding to specific target genes and utilizing feedback mechanisms that rely on histone modifications to form Polycomb chromatin domains (reviewed in [83]). These chromatin domains are characterized by H2AK119ub1, H3K27me3, and high-level occupancy of Polycomb group proteins to effectively repress transcription of target genes in differentiated somatic cells.

*Drosophila* GSCs contain non-canonical chromatin, with a moderately broad enrichment of repressive H3K27me3 in transcriptionally repressed genes and without the typical enrichment of H3K27me3 on Polycomb domains. The absence of high levels of H3K27me3 and PcG proteins within these domains is reminiscent of undifferentiated chromatin from *Drosophila* pre-MBT embryos and pre-implantation mouse embryos [86,87,88]. This suggests a conserved epigenetic ground state for the proliferation of undifferentiated precursor cells. Importantly, whereas the differentiation of nurse cells is linked to a “somatic-like” accumulation of H3K27me3 and PcG proteins within Polycomb domains [88], the oocyte preserves the characteristic wide distribution of H3K27me3 and PcG proteins [81,88].

The Pho repressive complex (PhoRC) binds to Polycomb Response Elements (PREs) within the DNA and, with Sex comb on midleg (Scm), helps to recruit Polycomb repressive complexes (PRC1 and PRC2) for transcriptional repression of neighboring genes [89,90,91,92]. PhoRC is composed of Pleiohomeotic (Pho) and Scm-related genes containing four mbt domains (Sfmbt) [89,90]. Polycomb-like (Pcl) and Scm regulate the transition of non-canonical to canonical chromatin during germ cell precursor differentiation [88]. Pcl is highly expressed in the germ cell precursors, where it inhibits PRC2 silencing and promotes GSCs’ non-canonical chromatin by changing PRC2 residence time on chromatin [88]. Depletion of Pcl within the female germ line changes the PRC2 chromatin binding footprint within the GSCs, with increased levels of H3K27me3 within the Polycomb domains [88].

However, the germ line requirement for different PcG subunits varies significantly [88,93,94,95]. Whereas E(z) and Su(z)12, two key subunits of PRC2, are essential to maintain oocyte fate and repress hundreds of genes in nurse cells [88,93,94], other Polycomb group proteins appear to regulate far fewer genes and have more subtle developmental phenotypes. For example, RNAi depletion or a null mutant allele for Polycomb (Pc), a conserved subunit of PRC1, only upregulates a small number of Polycomb target genes and does not show obvious oocyte or female fertility defects [88,94]. Depletion of PhoRC subunit Sfmbt has a significant impact on germline expression of a large number of genes, with defects in SC disassembly, meiosis, egg laying, and female fertility [95]. However, contrary to E(z) and Su(z)12, Sfmbt is not required for oocyte determination and maintenance of its fate [95].

As mitotic germ cell progenitors differentiate into oocytes and nurse cells, the expression of several progenitor genes needs to be repressed. While some genes are repressed by PhoRC, others are regulated by different transcription factors or chromatin-remodeling proteins. In some cases, both PhoRC and other regulators are involved in this repression. PRC2 activity is likely to enhance the activity of both types of repressors. All in all, this implies the existence of a multilayered regulatory mechanism that has likely evolved to achieve varying degrees of repression for different target genes during female gametogenesis.

SET Domain Bifurcated Histone Lysine Methyltransferase 1 (SetDB1) (also known as eggless) is required for deposition of repressive histone H3 lysine 9 trimethylation (H3K9me3) marks and heterochromatin formation [96,97]. SetDB1 is expressed in the germ line, mostly during early oogenesis [96]. In the GSCs and dividing cystocytes, it is equally expressed in the cytoplasm and in the nucleus, but after a 16-cell germline cyst, it accumulates mostly in the nucleus, more specifically, at the pericentric heterochromatin [96,97]. *SetDB1* mutants show significant GSCs differentiation defects [98], with the germaria lacking recognizable regions [96], without egg chamber formation and significant levels of apoptosis [97]. SetDB1 is also required for piRNA production and avoidance of transposons upregulation in the germ line and in the soma [98]. Transposon upregulation generates DNA damage in the germ line and most likely contributes to the GSCs differentiation defects of *Setdb1* mutants [98].

Stonewall (Stwl) is a protein that potentially interacts with SetDB1 [99] and associates with heterochromatin [100], and its depletion is associated with reduced levels of H3K9me3 and H3K27me3 [100]. Stwl is required within the germ line for GSC self-renewal [101], and ovaries mutant for *stwl* show misexpression of multiple genes, including the differentiation factor Bgcn, male germline sex-determination switch Phf7, and distinct testis-enriched genes [102].

#### 3.2.2. Active Histone Marks (H2B Monoubiquitylation H3K4me3, H3K36me3 and H3K79me3)

Bre1, an E3 ubiquitin ligase, mediates monoubiquitination of histone H2B and, indirectly, trimethylation of histone H3 lysine 4 and lysine 79 (respectively, H3K4me3 or H3K79me3), whereas Set1 is the main H3K4 tri-methyltransferase [103,104,105]. Bre1 and Set1 are required for normal levels of H3K4me3, and their function within the germ line or within the associated somatic cap cells is necessary for GSC self-renewal and avoidance of stem cell loss [106]. Bre1 and Set1 are similarly important within the somatic escort cells for cystoblast/cystocytes differentiation, as they limit BMP-signaling within the germarium [106]. Since Trithorax-related (Trr), another H3K4 tri-methyltransferase is similarly important for later stages of *Drosophila* oogenesis [107,108], this highlights the importance of H3K4me3 for different stages of female germ line development.

Male-specific lethal 3 (Msl3) and SET domain containing 2 (Set2) are also required for GSC differentiation [109,110]. Set2 encodes a methyltransferase that marks active genes with histone H3 lysine 36 trimethylation (H3K36me3) [111,112]. Set2 nuclear expression and the bulk levels of H3K36me3 increase in the differentiating cystoblasts, which suggests a role in the transcriptional activation of distinct differentiation factors [109]. Consistently, Set2 and Bam cooperate to promote cystoblast differentiation. Levels of nuclear Set2 and H3K36me3 are reduced in cystoblasts mutant for *bam*, whereas ectopic expression of bam increases the levels of H3K36me3 in the GSCs. Depletion of Set2 suppresses the GSC maintenance defects observed after ectopic expression of bam, which indicates that Set2 is downstream of Bam function during germ cell differentiation.

Msl3 preferentially binds nucleosomes marked with H3K36me3 [112]. Msl3 cooperates with histone acetyltransferase complex Ada2a-containing (ATAC) to regulate GSC differentiation [110]. Set2, Msl3, and Negative Cofactor 2β (NC2β) (an ATAC component) promote transcription of ribosomal protein S19 paralog (RpS19b), which is required for translation of Rbfox1, a key factor required for cystocytes differentiation and oocyte determination. Transcription of several SC components is similarly regulated by Set2, Msl3, and NC2β, but in this case, they do not require RpS19b or Rbfox1 translation.

Lysine demethylase 5 (Kdm5) (also known as little imaginal discs (lid)) encodes an H3K4 histone demethylase [113,114], capable of regulating transcription through both demethylase-dependent and demethylase-independent mechanisms. Loss of Kdm5 demethylase activity is associated with increased levels of H3K4me3 in the soma and in the germ line [81,115,116]. Counterintuitively, Kdm5 also interacts, through its PHD-domain, with H3K4me3-enriched chromatin [117], where it positively regulates transcription of distinct genes with high levels of active RNA polymerase II, H3K4me3, and H3K36me3 marks [115,117]. These observations illustrate the dual nature of Kdm5 as a context-specific transcriptional regulator capable of inducing or repressing gene expression. For example, depletion of Kdm5 is associated with reduced transcriptional activation of distinct target genes, including oxidative stress and mitochondria genes [115,117,118]. Kdm5 and Forkhead box, sub-group O (Foxo) cooperated for binding to promotors of co-regulated genes [118], clearly suggesting that Kdm5, in addition to its repressive demethylase activity, can also behave as a docking site of distinct transcription factors. Furthermore, suggesting an additional layer of regulatory complexity, Kdm5 interacts with the co-repressor Sin3 complex [119], which behaves as a scaffold protein of distinct histone deacetylases. Suggesting a functional relevance for this interaction, depletion of Kdm5 or Sin3 is associated with similar adult wing phenotypes and transcriptional misregulation (up-regulation or down-regulation) of a partially overlapping set of genes [119].

Loss of Kdm5 demethylase activity is associated with a major increase in the bulk levels of H3K4me3 within the female germ line [81,116]. Not surprisingly, Kdm5 has multiple functions during oogenesis, including the maintenance of the SC during pachytene, ensuring the correct architecture of the meiotic chromatin, avoiding precocious transcriptional reactivation of the oocyte during late prophase I, and sperm nuclear decompaction and karyogamy [81,116,120]. Whereas Kdm5 demethylase activity is not required for SC maintenance [116], it is, however, important for the expression of deadhead (dhd) and oocyte-to-zygote transition after fertilization [120]. Regarding the architecture of the oocyte meiotic chromatin, Kdm5 demethylase activity is required, at least partially, and when the experiment is performed in a *Kdm5* loss-of-function mutant background, for the correct organization of the karyosome [81], a compact cluster of meiotic chromosomes. Finally, and confirming the functional relevance of Kdm5 interaction with Sin3 for gene expression regulation, depletion of Sin3 is similarly associated with defects in dhd expression and an abnormal architecture of the meiotic chromatin [120,121].

#### 3.2.3. ATP-Dependent Chromatin-Remodeling

ATP-dependent chromatin-remodeling factor Imitation SWI (Iswi) and the histone H2B ubiquitin protease Scrawny (Scny) were also reported to be rate-limiting for GSC self-renewal [122,123]. Iswi is required for the maintenance of GSC because it facilitates BMP-signaling-mediated transcription [122]. More precisely, although the levels of pMad are normal in GSCs mutant for *Iswi*, the transcriptional response to such signaling is abnormal, with the precocious induction of Bam expression [122]. Scny encodes a hydrolase that deubiquitinates polyubiquitinated proteins [123]. GSC mutants for *scny* show elevated levels of ubiquitinylated H2B and H3K4me3. Interestingly, whereas histone H2B monoubiquitylation has been associated with RNA polymerase II elongation rates [124], Spt6, a transcriptional elongation factor, has been reported to be highly expressed in the GSCs and it is required for their self-renewal [125,126].

Further supporting the role of ATP-dependent chromatin remodeling complexes for GSC self-renewal, Brahma (Brm), the ATPase subunit of the *Drosophila* Swi/Snf chromatin-remodeling complexes, is also required for maintaining GSCs [127]. Germ line removal or depletion of Brm function is associated with the loss of the GSCs, without obvious defects of germ cell differentiation within the germarium.

## 4. *Drosophila* and Mammalian Oogenesis

Mouse germ line cysts differentiate into distinct cell types: one becoming the oocyte and the others becoming supporting nurse cells (reviewed in [2]) [6,7]. Similarly to *Drosophila*, mouse-supporting nurse cells transfer their cytoplasm and organelles into the presumptive oocytes before dying by non-apoptotic programmed cell death [6,7,128]. Multiple mouse oocytes can be determined within large cysts, with variable degrees of interconnections and fragmentation. However, after cyst fragmentation, typically, only one cell maintains an oocyte fate per cyst. Initially, all cells within the cyst express a similar transcriptome, as they are all likely to have entered meiosis [7]. Subsequently, most cells within each cyst are sequentially activated as nurse cells, most likely by the associated somatic pregranulosa cells, and the pro-oocytes start accumulating oocyte determinants from the interconnected nurse cells [7]. Suggesting some degree of conservation, *Drosophila*, and mouse oocytes are selected from the cystocyte with the highest number of intercellular bridges; they both acquire a large microtubule-organizing center (MTOC) and form a Balbiani body, which is an organelle comprised of mitochondria, endoplasmic reticulum, and RNAs [7]. The associated somatic cells also induce nurse cell turnover by non-cell autonomous programmed cell death [7]. Similarly to *Drosophila*, activated nurse cells also modulate the expression of distinct chromatin remodeling proteins, which is likely to influence mouse oocyte chromatin architecture and its differentiation [7].

During development, mammalian germ cells undergo significant epigenetic reprogramming, with global alterations in DNA methylation and histone modifications [129,130,131]. Levels of DNA methylation are reduced in migrating primordial germ cells (PGCs) and upon entry into the gonads [130], whereas the levels of histone H3 lysine 9 dimethylation (H3K9me2) and H3K27me3, respectively, decrease and increase [129,131]. H3K4me3 and histone H3 lysing 9 acetylation (H3K9Ac) levels in PGCs are early on similar to neighboring somatic cells, but they increase sharply as PGCs enter the genital ridge [131]. Levels of H3K9me3 are kept largely unchanged during this period of PGC development [131].

De novo DNA demethylation occurs in growing oocytes after reactivation of primordial follicles, and it is regulated by a complex interplay between the DNA methyltransferases DNMT3A and DNMT3L, and the histone marks H3K4me2/3 and H3K36me2/3 (reviewed in [132]). H3K4me3, PcG proteins, and maternally inherited Polycomb-mediated H3K27me3 also have a crucial role in the regulation of oocyte and early embryo chromatin architecture organization, as well as in post-implantation embryonic development (reviewed in [132]) [133,134,135]. As previously mentioned, these histone modifications play a crucial role in *Drosophila*, contributing to both the self-renewal and differentiation of GSCs, as well as the determination and differentiation of oocytes. Recently, it was also suggested that maternally inherited Polycomb-mediated H3K27me3 is required for a maternal-to-zygotic transition during *Drosophila* early embryonic development [136] and that histone H4 lysine 16 acetylation (H4K16Ac) is maintained throughout *Drosophila* and mammalian oogenesis and early embryonic development, to prime future gene activation [137]. However, significant epigenetic differences have also been reported. For example, *Drosophila* has low levels of DNA methylation [138], and there is no obvious requirement for PRC1 during *Drosophila* oogenesis [94].

## 5. Final Remarks and Conclusions

Although it is plausible that meiosis and female gametogenesis originated in a single eukaryotic ancestor, the degree to which the similarities observed in *Drosophila* and mammalian oogenesis reflect evolutionarily conserved mechanisms versus convergent evolution, with the rewiring of highly conserved regulatory modules remains uncertain. Ongoing work with various organisms representative of eukaryotic evolution (e.g., [10,139,140,141]) will undoubtedly contribute to tackling this problem and allow a better understanding of oogenesis evolution.

While mammalian in vitro gametogenesis has shown remarkable progress in replicating female gametogenesis [142,143], the continued use of invertebrate model organisms, such as *Drosophila* and *Caenorhabditis elegans*, can still provide valuable insights into the mechanisms of germ cell development and differentiation. For instance, by combining *Drosophila’s* exceptional genetics with ongoing research on distinct invertebrate model organisms (arthropod and non-arthropod), whose available genetic tools maybe be more limited, there is a unique opportunity for highly collaborative studies focused on gaining a molecular understanding of the evolution of female gametogenesis and its contribution to distinct reproductive strategies.

## Figures and Tables

**Figure 1 jdb-11-00021-f001:**
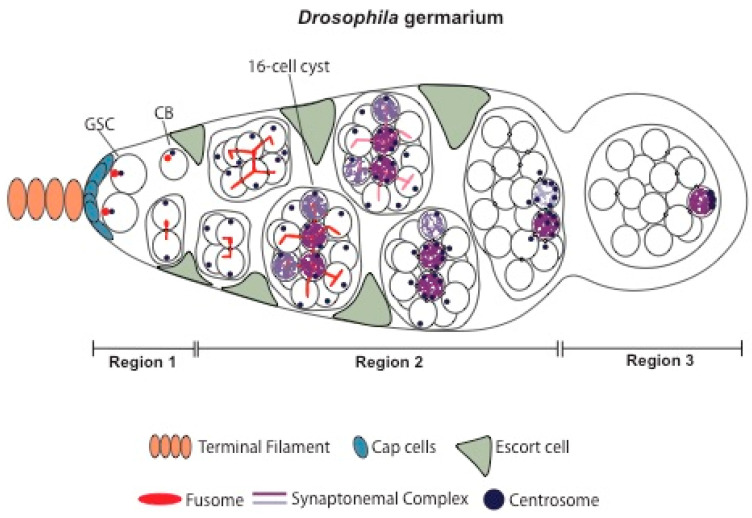
*Drosophila melanogaster* germ line cyst differentiation. *Drosophila* oogenesis starts in the germarium with an asymmetric division of an anteriorly localized germ line stem cell (GSC) and the formation of a cystoblast (CB) daughter cell. The germ line stem niche is composed of somatic terminal filament cells and cap cells that anchor the GSCs to the niche via adherens junctions and gap junctions and whose paracrine function is critical for the maintenance of the stem cell pool. GSCs’ mitotic divisions are typically asymmetric and orthogonal to the niche in such a way that the anteriorly localized daughter cell stays at the stem cell niche and maintains an undifferentiated stem cell fate, whereas the posterior localized daughter cell has the ability to exit the niche and initiate a highly complex differentiation program. The CB undergoes four rounds of mitotic divisions with incomplete cytokinesis, giving rise to a cyst with 16 cells interconnected by ring canals. Although interconnected, only one of these cystocytes (also known as cyst cells) will differentiate into the oocyte, and the other 15 cells will become supporting nurse cells. As the cyst is differentiating, more than one cystocyte will enter the meiotic program, assembling the synaptonemal complex (SC). This meiotic structure starts to be assembled in the cells with the highest number of ring canals, generally the ones with four- and three-ring canals, but will be soon disassembled, remaining the SC fully assembled only in the cell that will become an oocyte. The fusome is a dynamic cytoplasmic structure composed of cytoskeletal proteins. This structure is present initially as a small plug in the CB. As the cyst divides, the fusome is nucleated in each cell and fuses together, giving rise to a branched structure.

**Figure 2 jdb-11-00021-f002:**
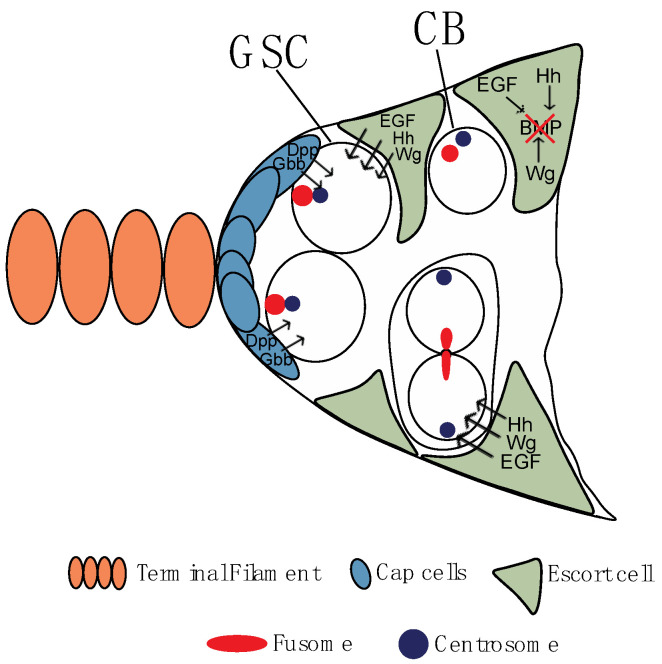
The germ line stem niche is composed of somatic terminal filament cells and cap cells that anchor the germ line stem cell (GSC) to the niche via adherens junctions and gap junctions and whose paracrine function is critical for the maintenance of the stem cell pool. The intermingled somatic escort cells also have a critical role in the regulation of GSC self-renewal, cystoblast mitotic divisions, and cystocyte differentiation, as they secrete distinct signaling molecules (e.g., Hedgehog (Hh), Epidermal Growth Factor (EGF), Wingless (Wg), insulin, and ecdysone) whose function is crucial for the regulation of early oogenesis.

**Figure 3 jdb-11-00021-f003:**
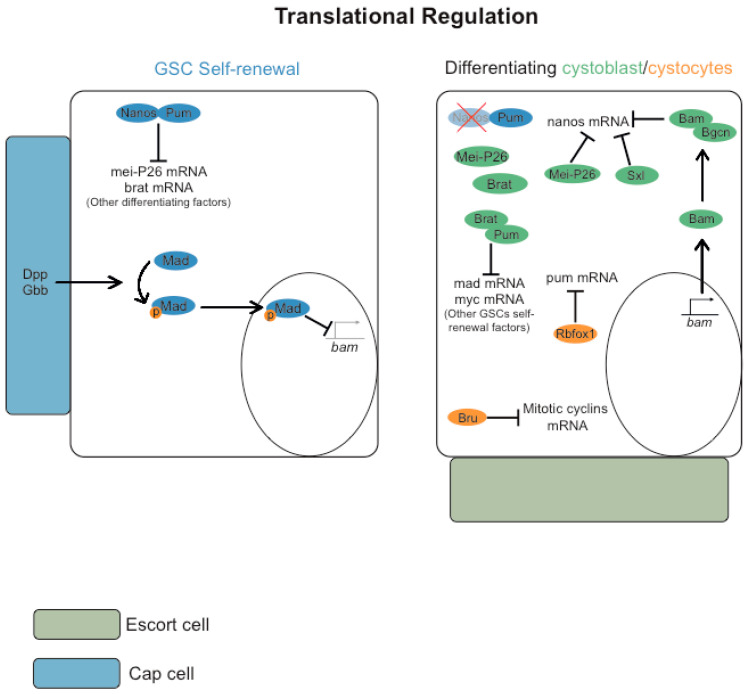
Translational regulation of GSC self-renewal and differentiation. GSC self-renewal and differentiation of the mitotically-dividing germ cell precursors are regulated by distinct regulators of translation. The translational repressor Pumilio (Pum), in conjunction with Nanos, is required for GSC self-renewal as they repress the translation of several differentiation factors, including Mei-P26 and Brain tumor (Brat). The transcription factor Mad is critical for GSC self-renewal, and after being phosphorylated by activated BMP receptors, it is translocated to the nucleus. Among other targets, pMad inhibits the transcription of differentiation factor bam. Bam, and its obligate co-factor Benign gonial cell neoplasm (Bgcn), form a translational repressor complex that antagonizes Nanos expression. The relationship between Pum, Nanos, and Brain tumor (Brat) is crucial for the switch from a GSCs self-maintenance program into the onset of cystoblast differentiation. Brat and Nanos compete for Pum binding, and the two translation repressor complexes Nanos-Pum and Brat-Pum target distinct subsets of mRNAs. Whereas the Nanos-Pum complex primarily represses the translation of distinct differentiation factors, such as Mei-P26 and Brat, Brat-Pum primarily represses the translation of GSC self-renewal factors, such as Mad and Myc. This generates a positive feedback loop for germ cell differentiation, where reduced levels of Nanos attenuate translation repression of brat mRNA, and Brat further competes with Nanos for Pum binding and enhances its own translation. In the eight-cell cyst, a switch from mitotic to meiotic programming is induced by the expression of RNA-binding Fox protein 1 (Rbfox1). Rbfox1 expression increases during early germ cell differentiation, and this RNA-binding protein directly represses the translation of pum mRNA. Bruno (bru), also known as arrest, encodes a translational repressor whose function is required to maintain mitotic quiescence during meiosis. Bruno represses the translation of mitotic cyclins during meiosis.

**Figure 4 jdb-11-00021-f004:**
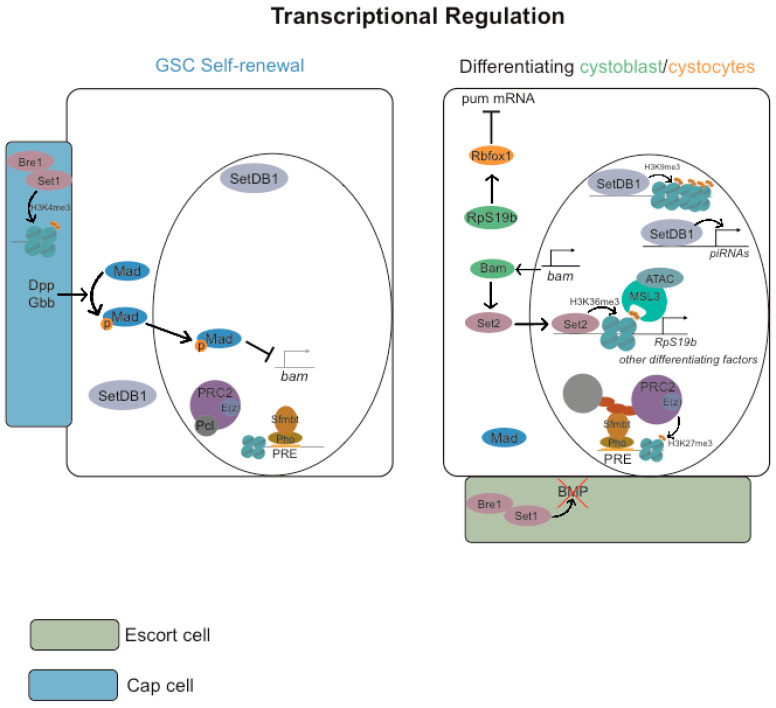
Transcriptional regulation of GSC self-renewal and differentiation. *Drosophila* GSCs contain non-canonical chromatin without the typical enrichment of H3K27me3 and high-level occupancy of Polycomb group (PcG) proteins on Polycomb chromatin domains. Polycomb-like (Pcl) and Sex comb on midleg (Scm) regulate the transition of non-canonical to canonical chromatin during germ cell precursors differentiation. Pcl is highly expressed in the germ cell precursors, where it inhibits PRC2 silencing and promotes GSCs’ non-canonical chromatin by changing PRC2 residence time on chromatin. SET Domain Bifurcated Histone Lysine Methyltransferase 1 (SetDB1) (also known as eggless) is required for deposition of repressive histone H3 lysine 9 trimethylation (H3K9me3) marks and heterochromatin formation. SetDB1 is required for GSCs differentiation and piRNA production. Bre1, an E3 ubiquitin ligase, mediates monoubiquitination of histone H2B and, indirectly, trimethylation of histone H3 lysine 4 and lysine 79 (respectively, H3K4me3 or H3K79me3), whereas Set1 is the main H3K4 tri-methyltransferase. Bre1 and Set1 are required for normal levels of H3K4me3, and their function within the germ line or within the associated somatic cap cells is necessary for GSC self-renewal and avoidance of stem cell loss. Bre1 and Set1 are similarly important within the somatic escort cells for cystoblast/cystocytes differentiation, as they limit BMP-signaling within the germarium. Set2 encodes a methyltransferase that marks active genes with histone H3 lysine 36 trimethylation (H3K36me3). Set2 nuclear expression and the levels of H3K36me3 increase in the differentiating cystoblasts, which suggests a role in the transcriptional activation of distinct differentiation factors. Consistently, Set2 and Bam cooperate to promote cystoblast differentiation. Msl3 preferentially binds nucleosomes marked with H3K36me3. Msl3 cooperates with histone acetyltransferase complex Ada2a-containing (ATAC) to regulate GSC differentiation. Set2, Msl3, and Negative Cofactor 2β (NC2β) (an ATAC component) promote the transcription of ribosomal protein S19 paralog (RpS19b), which is required for translation of Rbfox1, a key factor required for cystocytes differentiation and oocyte determination.

## Data Availability

Not applicable.

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
