# Peer review of "Genetic and Epigenetic Regulation of Drosophila Oocyte Determination"

_jdb, 2023, doi:10.3390/jdb11020021_

Round 1
Reviewer 1 Report
In this manuscript, Cabrita and Martinho summarized the current understanding and progress on Genetic and epigenetic regulation of Drosophila oocyte determination. The manuscript presents valuable insights, and I have identified some areas that could be improved prior to publication. The following points should be considered:
11. Niche Signal: It would be beneficial to include a figure illustrating the relationships among various pathways, such as BMP, Hedgehog, EGFR, and Wingless, to provide a visual representation of their interactions and the overall regulatory network. This visual aid would enhance the clarity and understanding of the niche signaling process.
22. Oocyte-Specific Factors: While the manuscript discusses proteins such as Orb and BicD, it would also be valuable to provide a detailed discussion on oocyte-specific mRNA factors, including Bicd and Osk. Highlighting the roles and regulation of these mRNA factors would offer a more comprehensive understanding of their contributions to oocyte determination.
33. Regarding the fusome, I recommend that it could include a more detailed discussion on its composition and function. Exploring the components and functions of the fusome in greater depth will enhance the significance and understanding of the fusome's role in the context of oocyte determination in Drosophila.
44. The establishment of a microtubule organizing center in the cyst is also essential for oocyte differentiation, so it is better to discuss it in the ms.
Reviewer 2 Report
The review summarizes the information on which genes, pathways and epigenetic factors in the syncytium, which has arosen as a result of incomplete mitotic divisions of the primary germ cell, contribute to the transformation of one of the cells into an oocyte while other syncytial cells enter the nurising pathway, providing pro-oocyte cytoplasmic substances and organelles. This review touches upon a topical subject-matter, written quite clearly, and provides a good reading. I do not have any major comments but still invite the authors to pay attention to some my considerations in order to more increase the value of their work.
Although the article primarily discusses data related to Drosophila oogenesis, yet from the beginning of the article, the authors emphasize a conservation between flies and mammals at the earliest stages of oocyte deternination, which occurs with the assistance of the nurse cells within a female germline cyst. This comparison is undoubtedly interesting and important, but I believe that the narrative might be slightly changed to make the review easier to percept by a wide range of readers — especially since the review already contains the special section 4 (Drosophila and mammalian oogenesis), which could be expanded by transferring references to mice from other places (e.g., lines 39-42). The main point is that while much has for a very long time been known about nurse cells and germline cysts in flies, the concept of mammalian nurse cells has appeared only in recent years through a series of highly impact papers by Professor Allan Spradling and his colleagues. At the same time, the Oxford Dictionary of Genetics still defines nurse cells as cells that nourish the oocyte in the insect ovary, generated by cystocyte divisions in the polytrophic meroistic ovary.
The paper begins (lines 22-23) with the phrase “Polyploidization and formation multinucleated structures...” followed by the sentence (lines 32-33) “cells within the Drosophila and mouse germ line cysts differentiate into distinct cell types”. Indeed, the nurse cells of insects with meroistic ovaries, including Drosophila, are highly polyploid, but nobody knows whether the nucleus of mouse nurse cells is polypoid due to endoreduplication of the genome to enhance RNA producing, as in flies. I am not sure if this true for mouse cystocytes (nurse cells) given their very short lifespan. Since ploidy of the nurse cells is not discussed elsewhere further, I recommend rewriting this part. Besides, it is not entirely clear at the first mention that “multinucleated structures” means germline cysts or cell clusters, since some “exotic” cases are known when incomplete cytokinesis leads to the formation of multinucleated oocytes, as in some frogs. Therefore, after the mentioned word-group, I would recommend to clarify that we are talking about germline cysts, providing their definition here (see lines 27-28). Further throughout the text, I do not recommend using the term “cyst cell”, since other cyst cells are known — e.g., the cells of somatic origin, which enclose spermatogenic cysts in the testes of some invertebrates. In the context of the present paper it seems to be preferable to say “cystocytes” only, although the term “cyst cell” is indeed used in relation to female germline cysts.
Additional minor points
All genus and species names must be italicized throughout the text (e.g., melanogaster, line 10 and so on throughout the text).
The word “golgi” (line 173) must be replaced with Golgi apparatus (Golgi is the surname of the scientist).
In my opinion, it would be useful for readers if the authors explained (defined) what a karyosome is and why its “the correct organization” is important for oocyte to develop (lines 450-451).
Reviewer 3 Report
The paper discusses the process of Drosophila oogenesis and highlights critical genes and pathways involved in determining and differentiating germ cells. It is well-written and addresses an important topic.
However, I have a few suggestions:
- The sentence in lines 25-27 needs improvement as it is unclear.
2. Multiple citations are missing throughout the article
Line 27
Line 36
Line 42
Line 146
Line 158
Line 168
Line 328
Line 331
Line 335
Line 383
Line 478
Line 482
Line 488
3. Line 162: The article mentions "Region 2A", but it is unclear what this refers to. It would be helpful to provide more information about this region for readers to understand.
4. Line 478: "However, and after cyst fragmentation…."
Remove the word "and"
5. Multiple instances in the text require Drosophila to be italicized. Eg, line 60, 102
